# Role of physical activity in the relationship between recovery from work and insomnia among early childhood education and care professionals: a cross-sectional study

Tiina Karihtala [ORCID] ,[1,2,3] Sampsa Puttonen,[2,4] Anu M Valtonen,[3] Hannu Kautiainen,[5,6] Leila Hopsu,[2] Ari Heinonen[1]

¹University of Jyväskylä Faculty of Sports and Health Sciences, Jyvaskyla, Finland
²Finnish Institute of Occupational Health, Helsinki, Finland
³Metropolia University of Applied Sciences, Helsinki, Finland
⁴Faculty of Social Sciences, University of Tampere, Tampere, Finland
⁵Primary Health Care Unit, Kuopio University Hospital, Kuopio, Finland
⁶Folkhälsan Research Center, Helsinki, Finland

**Correspondence to**
Tiina Karihtala;
tiina.karihtala@metropolia.fi

## ABSTRACT

**Objectives** This study aimed to investigate the association between recovery from work and insomnia and the role of objectively measured leisure-time physical activity and occupational physical activity in this association.

**Design** Cross-sectional.

**Setting and participants** Study with female early childhood education and care professionals (N=224) in Finland was conducted between April 2017 and September 2018.

**Methods** Recovery from work was measured with the Need for Recovery scale and insomnia with the Jenkins Sleep Scale. Physical activity was measured with an accelerometer for 7 days and analysed to represent leisure-time physical activity and occupational physical activity (min/day).

**Results** Both Jenkins Sleep Scale and occupational physical activity significantly predicted Need for Recovery (β=0.29; 95% CI 0.17 to 0.42 and β=0.14; 95% CI 0.01 to 0.27, respectively). A low relationship was observed between the Need for Recovery and Jenkins Sleep Scale (r=0.32, 95% CI 0.19 to 0.44). After categorising participants into four groups based on median splits of occupational and leisure-time physical activity, relationships between the Need for Recovery and Jenkins Sleep Scale were low to moderate in the high occupational physical activity and leisure time physical activity group (r=0.38, 95% CI 0.14 to 0.61), and in the high occupational physical activity and low leisure-time physical activity group (r=0.40, 95% CI 0.18 to 0.63).

**Conclusion** Both insomnia and physical activity at work seem to be relevant in recovery from work. To enhance recovery, especially those involved in high physical activity at work, should seek methods to improve recovery, by incorporating activities that promote recuperation both during their workday and in their leisure time. Further research on the relevance of physical activity in recovery with longitudinal setting is warranted.

**Trial registration number** NCT03854877.

## BACKGROUND

Changes in working life, such as increased work intensity and difficulty of separating work from leisure time, challenge relaxation

### STRENGTHS AND LIMITATIONS OF THIS STUDY

⇒ In this study, an accelerometer was used as an objective tool to measure physical activity.
⇒ Participants represent well early childhood education and care workers in Finland.
⇒ The study was cross-sectional, and longitudinal studies are needed to examine the temporal relationships.
⇒ The findings of this study may not be directly applicable to other occupations or to men.

and recovery after working hours. Recovery from work is essential for an individual's work ability and health,[1 2] and more attention should be drawn to the activities during non-work time to unwind from work-induced stress.[3] Successful recovery ensures sufficient energy for the next day's work activities and for meaningful free time or off-work responsibilities. On the other hand, lack of recovery can decrease work ability and increase the risk of early exit from work life.[1 4] The experience of insufficient recuperation from work and the need to take a break from ongoing activity is referred to as 'Need for Recovery (NFR) from work'.[5 6]

Demerouti *et al*[7] have suggested that the success of recuperating from workload is affected by the demands and resources both at work and at home and also by sleep and leisure-time activities. Sufficient and high-quality sleep enables brain and body to recover from physiological and psychological workload.[8 9] In contrast, insufficient sleep and insomnia have diverse health-deteriorating effects.[10–14] In addition, poor sleep quality decreases work ability[15] and is linked to increased sickness absences.[16] Yet, the association between insomnia and recovery from

work has received little attention. Preliminary evidence suggests that insomnia is associated with higher NFR from work.[17 18]

Karihtala *et al*[19] and Stevens *et al*[4] showed that a high level of accelerometer-measured occupational physical activity (OPA) is associated with increased NFR from work among early childhood education and care (ECEC) professionals and blue-collar workers. Contradictory results have been reported for office employees and healthcare workers in studies with self-reported physical activity.[20 21] On the other hand, leisure-time physical activity (LTPA) has been suggested to enhance recovery from work by supporting detachment from work.[22 23] However, we did not observe this among ECEC professionals.[19] To reduce discrepancies between studies and increase the reliability of the results, methodologies for physical activity measures should be unified; recommendations to implement objective measures have been presented.[24 25]

Systematic reviews have found positive associations between LTPA and sleep quality, sleep-onset latency and sleep duration.[26–28] However, the association between LTPA and insomnia was not detected in two other recent systematic reviews.[29 30] Regarding OPA, studies have found that a high level of physical activity during working hours is related to insomnia[31–35] and that LTPA may protect against insomnia even when high OPA exists.[33] Since the effects of OPA and LTPA on insomnia seem to diverge, it is important to examine physical activity in these two domains separately.

In supporting employee well-being, it is important to understand how on-work and off-work activities, including sleep, can promote recovery. So far, little research exists on the association between recovery from work and insomnia, and, to our knowledge, research among ECEC professionals in this regard is non-existing. Also, the role of physical activity during work or leisure time in this context is not yet fully understood. The aims of this study were, first, to investigate the association between recovery from work and insomnia, and, second, to study the roles of accelerometer-measured LTPA and OPA in this association in female ECEC professionals. Our hypothesis was that insomnia is associated with NFR from work, that LTPA would enhance recovery and that OPA would deteriorate it.

## METHODS
### Design and participants
This cross-sectional study is a secondary analysis of the DagisWork study, a randomised controlled trial (Workplace healthcare interventions to promote the work ability of kindergarten personnel, NCT03854877). It was conducted during 2017–2018 in 23 ECEC centres in 2 southern Finland cities. Study profile is presented in figure 1. Centres with less than seven workers were excluded due to the estimated excessive need of time and resources for baseline measurements. Exclusion criteria for a single participant were retirement or termination of employment within next 6 months or pregnancy. Men were excluded because of their limited number (n=3). Finally, a random sample of 23 centres with altogether eligible 386 employees was selected to the study. Hence, 117 individuals did not

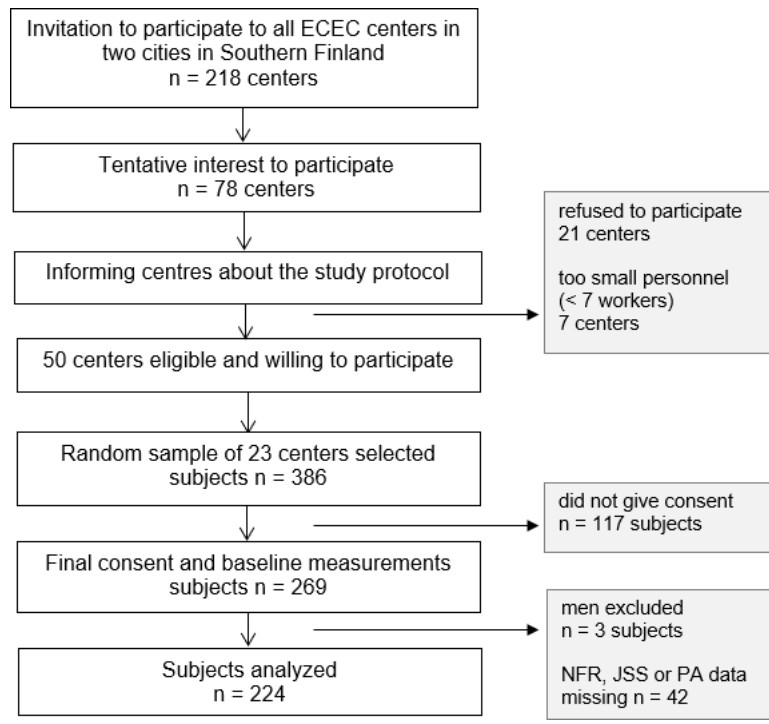

**Figure 1** Study profile. ECEC, early childhood education and care; JSS, Jenkins Sleep Scale; NFR, Need for Recovery; PA, physical activity.

give consent and 269 participants continued with baseline measurements (response rate 70%). Participants were not involved in designing, conducting, reporting or dissemination plans of this study. At the workplace, two healthcare professionals performed baseline measurements (weight, height and body mass index) and instructed participants on the use of a waist-worn accelerometer for the following week. Concurrent filling in a diary as well as an electronic questionnaire was guided. The 224 participants with complete data for recovery from work, insomnia and physical activity at baseline formed the final analytical sample.

### Recovery from work

Recovery from work was assessed with the NFR scale, which is known to be valid and reliable in assessing the need to recuperate from work (intra-class correlation coefficients [ICCs] 0.68–0.80, effect size 0.40).[6 36] Items in the scale represent reduced performance, feeling of overload, irritability, social withdrawal and lack of energy for new tasks, with statements such as the following: 'I find it difficult to relax at the end of a working day', 'I find it difficult to concentrate in my free time after work' or 'When I go home from work, I need to be left in peace for a while'. The final score of 0–100 was composed of 'yes' or 'no' answers in 11 dichotomous questions and was calculated as a percentage of positive answers. The higher the percentage score was, the higher was NFR from work. The scale was implemented in the electronic questionnaire, and at least 8 answers out of 11 were required in order to be included in the analysis.

### Insomnia

Insomnia symptoms were assessed using the Jenkins Sleep Scale (JSS),[37] a widely used tool to assess insomnia symptoms in the working-age population[38 39] with good psychometric properties in the Finnish version.[40] Participants were asked about difficulties falling asleep, waking up several times per night, difficulties staying asleep and non-restorative sleep (ie, feeling tired and worn-out waking up after the usual amount of sleep) during the previous month. The sum score of 4–24 comprised from the responses given on the scale, from 1 to 6 (1=not at all, 2=1–3 nights/month, 3=1 night/week, 4=2–4 nights/week, 5=5–6 nights/week, 6=every night).

### Physical activity

Physical activity was measured with the waist-worn accelerometer (ActiGraph GT9X Link, ActiGraph, USA) for seven consecutive days and nights. During the nights, the metre was worn in the wrist. The metre was only removed for showering or other water-related activities. At the same time, participants recorded working hours and sleep times in a diary. Awake time was separated into occupational and leisure time. Participants with data for at least 4 days and 10 hours/day were eligible for the final analysis. ActiLife software (V.6.13.3) was used for data analysis, and 30 Hz frequency and 60 s epochs were used. Non-wear time criteria from Choi *et al*[41] were applied, and a minimum length of 30 min and a drop time of 2 min for sedentary time were used. Cut points for different thresholds for physical activity intensity levels were used as follows: sedentary 0–99 counts per minute (cpm), light 100–1951 cpm, moderate 1952–5724 cpm, vigorous 5725–9498 cpm and very vigorous over 9499 cpm.[42] Counts per minute refer to the number of times that an accelerometer detects movement (change in acceleration). Sedentary time was excluded from data and average PA minutes/day were calculated for each PA intensity levels. For analysis, physical activity was defined as any level of physical activity (light, moderate, vigorous or very vigorous) and mean physical activity minutes/day at all PA intensity levels were added up. With the help of diaries, PA minutes were calculated separately for leisure time to represent LTPA and for occupational time to represent OPA. Later in this report high time (=high min/day) spent on LTPA or OPA will be referred as high LTPA or OPA.

### Other variables

The electronic questionnaire was used to register demographics and other characteristics of the participants, such as marital status, smoking (yes/no), alcohol consumption (units/week), educational level (university degree/vocational degree) and ECEC work experience (years).

Work ability was reported with a single question: 'How good is your current work ability compared with your lifetime best?' and was rated on a scale from 0 to 10 (0=completely unable to work; 10=work ability at its best).[43]

Mental health symptoms were measured with the General Health Questionnaire -12.[44] The 12 questions with response choices being 'much more than usual', 'rather more than usual', 'no more than usual' and 'not at all', were scored from 0 to 3, respectively. A sum score was calculated (0–36).

Perceived health was examined with one question: 'In general, would you say your health is?' Answers were given on a scale from 1=poor to 5=excellent, and values 4 and 5 were classified as 'good health'.[45]

Stress level was measured with the Perceived Stress Scale including 10 questions such as 'In the last month, how often have you felt nervous and stressed?' rated on a 5-point scale (0–4) and added up as a sum score of 0–40.[46]

Musculoskeletal, cardiovascular, respiratory or mental disorders were registered with open-response questions and dichotomised to yes or no, with the classification 'yes' requiring both diagnosis by a doctor and symptoms being present at the moment or occurring often or repeatedly.

**Table 1** Correlations between the Need for Recovery and individual insomnia symptoms measured by the Jenkins Sleep Scale

|  | r | 95% CI* |
|---|---|---|
| Difficulty to fall asleep | 0.17 | 0.04 to 0.30 |
| Waking up several times per night | 0.14 | 0.00 to 0.27 |
| Difficulty to stay asleep | 0.26 | 0.13 to 0.38 |
| Non-restorative sleep | 0.38 | 0.26 to 0.49 |

*CIs adjusted following the Sidak method.

Pain intensity during the last month was registered on 1–6 scale, from no pain to very severe pain, and pain interference on a 1–5 scale, from no pain to daily pain. Both scales were obtained from the Short Form 36 Health Survey Questionnaire pain[45] and were converted to a score of 0–100.

### Statistical analysis
Data are expressed as mean and SD or frequencies with percentages. The relationship between NFR and JSS was modelled using linear regression analysis. Results were analysed using factorial (two between-subjects factors: LTPA and OPA) analysis of variance and logistic models. Models included main effects of LTPA and OPA and their interaction. Multivariate linear regression analysis was used to identify the relationship between LTPA, OPA, JSS and age as continuous variables and the NFR with standardised regression coefficient beta. The beta value is a measure of how strongly the predictor variable influences the criterion variable. The beta is measured in units of SD. Cohen's standard for beta values above 0.10, 0.30 and 0.50 represents small, moderate and large relationships, respectively. Unadjusted and adjusted (partial) correlations were calculated by the Pearson method, using Sidak-adjusted CI. Stata V.17.0 (StataCorp) statistical package was used for the analysis.

### Patient and public involvement
None.

## RESULTS
The mean age of all participants was 44 (SD 11) years, and 65% of them had been working more than 10 years in an ECEC centre. They represented the typical employee groups in Finnish ECEC centres with managers (7%), special education teachers (6%), teachers (36%), child carers (42%) and assistants (9%). Participants' average NFR score was 37.5% (SD 26.3), and 30% of participants were classified as having high NFR from work (>54.5%). The mean JSS score was 6.7 (SD 4.3), and 10% of participants suffered from insomnia disorder (any insomnia symptom ≥5 nights /week). The mean time for LTPA was 252 min/ day (SD 66) and for OPA 228 min/day (SD 51).

### Association between recovery from work and insomnia in all participants
A low relationship was observed between NFR and JSS sum score (r=0.32, 95% CI 0.19 to 0.44). Table 1 shows the relationships between NFR and individual JSS symptoms including waking up several times per night, difficulty staying asleep and non-restorative sleep.

### Background characteristics
Participants were divided into four physical activity groups according to median values of LTPA (min/day) and OPA (min/day) (figure 2).

Participant characteristics are presented in these groups in table 2, showing how physical activity is reflected in the characteristics of different PA groups. No interactions between LTPA and OPA were detected in any of the variables. The results showed that high LTPA was related with a longer career history at an ECEC centre and low OPA to a higher level of education.

### Relationship between recovery from work and insomnia in four physical activity groups
Age-adjusted correlations between NFR and JSS for the high and low LTPA and OPA groups are presented in figure 3. Relationships between NFR and JSS were low to moderate in the high OPA and LTPA group (r=0.38, 95% CI 0.14 to 0.61), and the high OPA and low LTPA group (r=0.40, 95% CI 0.18 to 0.63). Correlations were low in the low OPA and LTPA group (r=0.20, 95% CI −0.02 to 0.43), and in the low OPA and high LTPA group (r=0.29, 95% CI 0.02 to 0.51). Hence, results indicate that

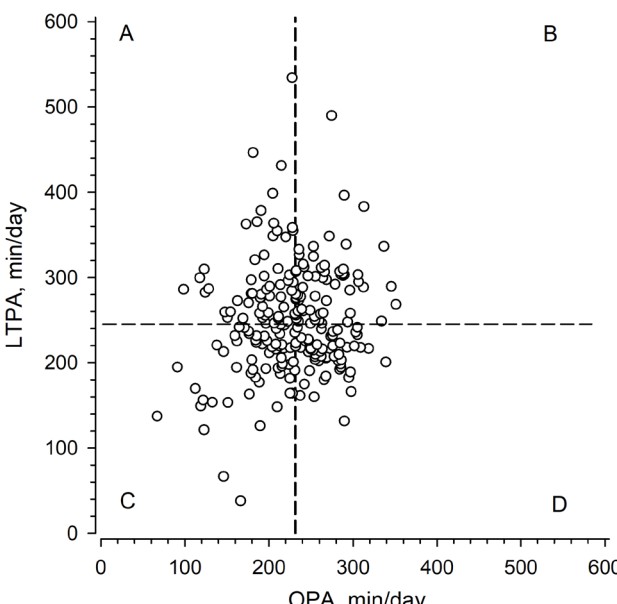

**Figure 2** Daily minutes of physical activity categorised into four groups based on median splits of leisure-time physical activity (LTPA min/day) and occupational physical activity (OPA min/day): (A) high LTPA+low OPA, (B) high LTPA+high OPA, (C) low LTPA+low OPA and (D) low LTPA+high OPA. Medians for LTPA (246.2 min/day) and for OPA (231.0 min/day) are illustrated as dashed lines.

**Table 2** Demographic and clinical characteristics of the participants according to low or high leisure-time physical activity (LTPA, min/day) and occupational physical activity (OPA, min/day)

| | High LTPA | | Low LTPA | | P value | | |
| --- | --- | --- | --- | --- | --- | --- | --- |
| | Low OPA | High OPA | Low OPA | High OPA | Main effect | | |
| | n=53 | n=59 | n=58 | n=54 | LTPA | OPA | Interaction |
| Age, mean (SD) years | 44 (9) | 45 (11) | 44 (11) | 43 (12) | 0.57 | 0.95 | 0.60 |
| In partnership, n (%) | 36 (71) | 40 (75) | 39 (70) | 38 (73) | 0.36 | 0.83 | 0.90 |
| Smoker, n (%) | 7 (14) | 11 (21) | 8 (14) | 9 (17) | 0.81 | 0.33 | 0.72 |
| Alcohol use*, mean (SD) | 2.0 (2.5) | 2.7 (3.6) | 2.1 (3.2) | 1.6 (2.1) | 0.22 | 0.72 | 0.15 |
| Highly educated, n (%) | 29 (57) | 21 (40) | 31 (56) | 14 (27) | 0.30 | <0.001 | 0.33 |
| Career history in an ECEC centre (years) | | | | | 0.009 | 0.51 | 0.37 |
| <1 | 2 (4) | 2 (4) | 0 (0) | 2 (4) | | | |
| 1–3 | 2 (4) | 5 (9) | 7 (13) | 8 (15) | | | |
| 4–10 | 10 (20) | 6 (11) | 15 (27) | 16 (31) | | | |
| >10 | 37 (73) | 40 (75) | 34 (61) | 26 (50) | | | |
| BMI, mean (SD) | 27.3 (7.8) | 27.0 (5.2) | 26.7 (6.1) | 27.5 (5.6) | 0.99 | 0.72 | 0.54 |
| Work ability, mean (SD) | 8.0 (1.3) | 8.2 (1.1) | 8.1 (1.3) | 8.4 (0.9) | 0.49 | 0.19 | 0.78 |
| GHQ-12, mean (SD) | 12.4 (5.6) | 12.5 (5.0) | 12.0 (5.0) | 11.2 (4.0) | 0.21 | 0.63 | 0.48 |
| Good health, self-rated, n (%) | 33 (65) | 35 (66) | 37 (66) | 29 (56) | 0.51 | 0.51 | 0.39 |
| Perceived Stress Scale, mean (SD) | 16.7 (6.5) | 16.3 (6.6) | 16.9 (6.2) | 14.8 (6.4) | 0.45 | 0.14 | 0.34 |
| Disorders, n (%) | | | | | | | |
| Musculoskeletal | 14 (27) | 17 (32) | 14 (25) | 8 (16) | 0.11 | 0.58 | 0.22 |
| Cardiovascular | 9 (18) | 8 (15) | 9 (16) | 7 (14) | 0.77 | 0.63 | 0.99 |
| Respiratory | 6 (12) | 6 (11) | 9 (16) | 6 (12) | 0.63 | 0.63 | 0.70 |
| Mental | 1 (2) | 5 (9) | 7 (13) | 3 (6) | 0.27 | 0.53 | 0.06 |
| Pain (SF-36), mean (SD) | | | | | | | |
| Intensity | 33 (23) | 32 (22) | 33 (22) | 33 (24) | 0.93 | 0.96 | 0.92 |
| Interference | 25 (21) | 24 (26) | 25 (25) | 23 (25) | 0.78 | 0.67 | 0.88 |
| OPA min/day, mean (SD) | 190 (31) | 267 (32) | 184 (37) | 269 (25) | – | – | – |
| LTPA min/day, mean (SD) | 305 (57) | 295 (41) | 196 (42) | 212 (23) | – | – | – |
| Need for Recovery mean (SD) | 35.8 (28.3) | 41.5 (28.9) | 37.4 (23.6) | 34.9 (24.4) | 0.50 | 0.66 | 0.27 |
| Jenkins Sleep Scale, mean (SD) | 6.8 (4.3) | 6.3 (4.1) | 6.8 (3.9) | 6.7 (5.0) | 0.70 | 0.59 | 0.65 |
| Difficulty falling asleep | 0.7 (1.0) | 0.9 (1.0) | 1.3 (1.2) | 1.0 (1.2) | | | |
| Waking up several times per night | 2.4 (1.6) | 2.5 (1.5) | 2.4 (1.6) | 2.3 (1.7) | | | |
| Difficulty staying asleep | 1.7 (1.6) | 1.8 (1.5) | 1.4 (1.4) | 1.7 (1.6) | | | |
| Non-restorative sleep | 2.3 (1.4) | 1.8 (1.3) | 2.0 (1.3) | 2.2 (1.4) | | | |

*Units per week (1 unit=12 g of pure alcohol), highly educated=university level degree.
BMI, body mass index; ECEC, early childhood education and care; GHQ-12, General Health Questionnaire-12; SF-36, Short Form 36 Health Survey Questionnaire.

NFR is related to insomnia, especially if physical activity during the working day is high.

### Relationship between recovery from work, physical activity and insomnia

Multivariate relationships between NFR and LTPA, OPA, age and JSS are presented in figure 4. Both JSS and OPA significantly predicted NFR, but the standardised regression coefficients were small (β=0.29; 95% CI 0.17 to 0.42 and β=0.14; 95% CI 0.01 to 0.27, respectively).

## DISCUSSION

We investigated the relationship between the recovery from work and insomnia, exploring whether physical activity has role on this relation in female ECEC professionals. Findings suggest that there is a relation between recovery from work and both insomnia and physical activity during working hours. The results supported our hypothesis regarding the detrimental relation of OPA on recovery, as recovery from work and insomnia were particularly associated with the groups engaged in higher physical activity during work, compared

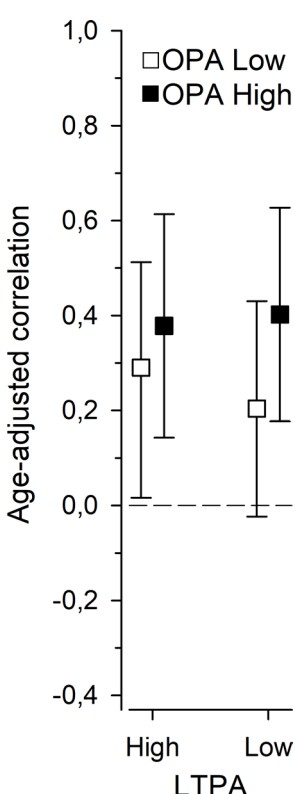

**Figure 3** Age-adjusted correlations with 95% CIs between Need for Recovery and Jenkins Sleep Scale for the physical activity (PA) groups based on low or high leisure-time physical activity (LTPA, min/day) and occupational physical activity (OPA, min/day): high LTPA+low OPA, high LTPA+high OPA, low LTPA+low OPA or low LTPA+high OPA.

with those with lower work-related activity. However, contrary to our hypothesis, the findings did not support the idea of recovery enhancing role of LTPA.

Average NFR among the female ECEC professionals was 38%. This is slightly higher than in airline workers (males 20% and females 28%), but substantially lower than in emergency department staff (82%).[47 48] We found that 30% of ECEC professionals had high NFR (>54.5%), similar to earlier studies with nurses and kindergarten teachers (30%) and industry and healthcare workers (21%).[21 49] In our

study, JSS mean score was similar to a large cohort study in Finland.[40] Occasional insomnia symptoms (any insomnia symptom 1–4 nights per month) were reported by 46% and insomnia disorder (any insomnia symptom ≥5 nights per week) by 10% of participants; the readings are very similar to those of a previous Finnish study (45% and 10%, respectively).[50] ECEC professionals spent 58% of their waking time sedentary and 42% in physical activity of which almost 90% was in light intensity. Proportions are very similar to earlier studies reporting accelerometer-derived physical activity levels in adult populations.[51 52]

Our results on relation between recovery from work and insomnia are in line with previous studies on this topic, where the population was slightly younger and, unlike our study, included both women and men.[16 17] Recovery from work associated with several insomnia symptoms (waking up several times per night, difficulties staying asleep and non-restorative sleep) reflecting diverse deficiencies with sleep health that might challenge recovery. Sleep disturbances have associated with sickness absence and even one night of sleep deprivation increases the need for recovery for the following days.[16 53] Hence, it might be valuable to explore ways to support unwinding from work already at an early stage of perceived NFR, in order to prevent insomnia and further to promote recovery. However, only few studies have explored the association between recovery from work and insomnia. More studies focusing on sleep health and recuperation from work in promoting recovery are warranted. Also, longitudinal studies are needed to examine cause-and-effect relationships in more detail.

We found that, in addition to insomnia, physical activity during working hours was associated with the need to recover from work. Results also suggested that if OPA was high the relationship between NFR and JSS was low to moderate, and if OPA was low the relationship was low or non-existing. These results support our hypothesis that high OPA could deteriorate recovery. It indicates that even low to moderate ECEC centre workload may be relevant in this context. Nevertheless, the findings need to be interpreted cautiously. The result may be due to the poor physical condition of the participants as physically

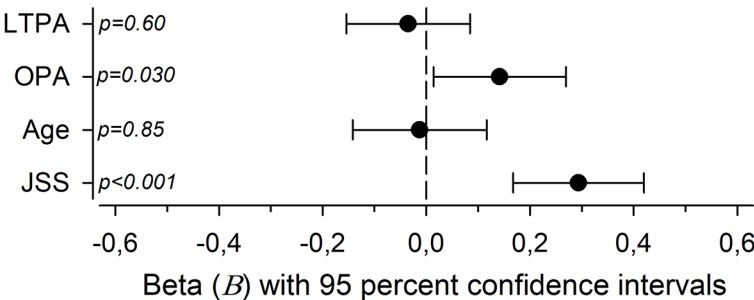

**Figure 4** Multivariate relationships between Need for Recovery (NFR) and occupational physical activity (OPA) level, leisure-time physical activity (LTPA) level, age and Jenkins Sleep Scale (JSS) sum score. Cohen's standards for beta values above 0.10, 0.30 and 0.50 represent small, moderate and large effect sizes, respectively.

demanding work and sleep difficulties have been found to associate with poorer physical function.[54 55] We did not measure participants' physical function; therefore, it remains unclear whether even a light workload, especially if long-lasting, may be too wearing and increase the NFR from work when one's physical condition is weak. Also, we noticed that low OPA was related to a higher level of education of participants. Employees in ECEC centres, who engage in higher levels of physical activity during work hours often hold roles that involve persistent interaction with children throughout the day, in comparison to their more educated counterparts. This job characteristic with active engagement and responsibility for children may intensify the perceived need for recuperation after working hours. OPA did not relate to other characteristics of participants. Also, we investigated only women, and they all worked in the same occupational setting (ie, an ECEC centre). The research should be repeated with men, as well as with different professions, including those with more physically demanding work tasks. In addition, there is a potential limitation in representativeness of the study sample as this study is a secondary analysis from the DagisWork study, a randomised controlled intervention trial and we used a baseline data for this analysis.

The results suggest that LTPA is not relevant to the relationship between insomnia and recovery from work. In addition, the career history in ECEC centre was related to LTPA, although age was not. High LTPA has been linked to better sleep[27 56] while the appropriate duration and intensity of physical activity are still debated.[27] It has also been suggested that a high LTPA may protect from insomnia even when activity during the working day is high.[33] However, this was not confirmed by our results and our hypothesis of recovery enhancing role of LTPA was not affirmed by the data. In some studies, high LTPA has been linked to a lower NFR.[20 57] Theoretically, we can assume that LTPA improves physical fitness, which increases stress tolerance and thereby reduces the NFR from work. On the other hand, LTPA may facilitate relaxation, promote psychological detachment from work and enhance experience of having autonomy and control over leisure-time choices and activities.[3] However, our results did not support these assumptions. First, this may be due to the fact that accelerometer-derived LTPA inadequately captured participants' leisure-time activities performed in stationary positions, such as strength training. Second, an accelerometer may not be able to capture activities unwinding from work, such as relaxation exercises or Pilates-type training, for example. Third, the intensity of LTPA among ECEC workers may be too low to improve physical fitness and thereby increase load tolerance at work. They spent 58% of their waking time sedentary, and only 5% in moderate or vigorous physical activity, similar proportions as a recent large cohort study reported.[58] Also, the only characteristic related to LTPA was the career history in ECEC centre, which we cannot explain, especially when age was not one of those factors. Further, what forms of exercise would best enhance recovery from work remains unknown. In the future, research should consider different domains of LTPA suitable at an individual level to enhance unwinding from work. For that, diary or similar methods would be needed parallel to technological measures. Recovery is a complex phenomenon that encompasses several facets of an individual's life, including their lifestyle, work and various aspects related to the life circumstances. Significance of physical activity in this context remains an intriguing subject of study, exploring the type, amount and intensity of physical activity that supports recovery for individuals.

## CONCLUSIONS

In conclusion, the results of this study suggest that both insomnia and physical activity during a working day are associated with recovery from work in ECEC professionals. Insomnia and recovery from work seem to relate especially if physical activity is high during a working day. This implies that ECEC professionals, especially those involved in high physical activity at work, should seek methods to improve their recovery, by incorporating activities that promote recuperation both during their workday and in their leisure time. Further research, particularly longitudinal studies, are warranted to explore the significance of both OPA and LTPA separately as part of the individual and multifaceted entity of recovery.

**Contributors** TK, SP, AMV, HK and AH were responsible for constructing an idea, research question and hypothesis for the study. TK and LH collected the data. Initial data processing was done by TK and HK was responsible for final statistics and figures. TK was responsible for initial manuscript and SP, AMV, HK, LH and AH have read, critically revised and approved the final version of this manuscript. TK is responsible for the overall content as guarantor.

**Funding** This work was supported by the Academy of Finland (287195), University of Jyväskylä, Faculty of Sport and Health Sciences (grant no: NA), Otto A. Malm Foundation, Helsinki, Finland (grant no: NA).

**Competing interests** None declared.

**Patient and public involvement** Patients and/or the public were not involved in the design, or conduct, or reporting, or dissemination plans of this research.

**Patient consent for publication** Not applicable.

**Ethics approval** This study involves human participants and was approved by Helsinki University Central Hospital coordinating ethics committee HUCH/1883/2016. Participants gave informed consent to participate in the study before taking part.

**Provenance and peer review** Not commissioned; externally peer reviewed.

**Data availability statement** Data are available on reasonable request. Availability with the permission of the Finnish Institute of Occupational Health.

**ORCID iD**

Tiina Karihtala http://orcid.org/0000-0002-4353-0955

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
