## [Reviewer comments · BMJ Open]

ARTICLE DETAILS

TITLE (PROVISIONAL)	Role of physical activity in the relationship between recovery from work and insomnia among early childhood education and care professionals: a cross-sectional study
AUTHORS	Karihtala, Tiina; Puttonen, Sampsa; Valtonen, Anu; Kautiainen, Hannu; Hopsu, Leila; Heinonen, Ari

VERSION 1 – REVIEW

REVIEWER	Lucion Loreto, Bibiana Bolten Universidade Federal do Rio Grande do Sul, Programa de Pós-Graduação em Psiquiatria e Ciências do Comportamento
REVIEW RETURNED	04-Oct-2023

GENERAL COMMENTS	I suggest correcting the following words on line 18 of page 6: "understood" instead of "understanded" and "firstly" instead of "first".
---

REVIEWER	Wentz, Kerstin University of Gothenburg
REVIEW RETURNED	17-Oct-2023

GENERAL COMMENTS	This is very valuable work and the research questions are very important and therefrom the results very interesting. However, the hypothesis should be developed concerning high and low OPA. Concerning the result section it needs to be less wordy, more structured and more informative e.g. picturing the results from Figure 1 and Table 1 respectively and then form a conclusion. The paragraph on Need for recovery should refer to table 1 etc. The result section also does not deal with the hypothesis on insomnia, need for recovery, OPA and LTPA which is necessary. The discussion also needs to be less wordy, more structured, selective, more distinct and more informative and include components regarding breaks and physical load during work. The references should be presented so that they foremost could shed light on the current result. Concerning the conclusion there is room for some developments. Degrees of freedom concerning breaks at work or reasonably balanced demands at work are not mentioned but solely physical activity as the source of stress tolerance, fitness and health.
--

REVIEWER	Bieleman, Andre Saxion University of Applied Sciences, Smart Health
REVIEW RETURNED	13-Nov-2023

GENERAL COMMENTS	GENERAL: The physiological perspective (by measuring physical activity) on need for recovery and sleep in this study is interesting. However, I
--

believe that the results should be interpreted with more constraints than the authors do, considering the low to moderate correlations found.

ABSTRACT:

In the conclusion the phrase 'insomnia and need for recovery are suggested to relate ...' is vague. The recommended 'attention' for phys.act. during free time is also vague and not supported by results.

INTRODUCTION:

Based on a good overview of relevant literature, the authors formulate a hypothesis for their study.

METHODS:

The background of the DagitWork study is described; complete data of 224 participants were analysed.

Please report brief information on psychometric aspects of the NFR. Please report brief information on the thresholds for physical activity, what do the counts per minute express exactly?

Please give brief explanation why these other variables (page 8, line 22) were selected and how they were included in analysis of data. They don't come back in the discussion.

Stat.analysis: what are FMI and LMI?

In Methods, the calculation of correlations coefficients for the 4 combinations of OPTA and LPTA is not mentioned or explained; no rationale for this approach is given.

RESULTS

Table 1:

What calculation do the percentages in the row Highly educate indicate? 29 is 57% of what (57% of 53 is 30)? The highly educated participants seem to have lower OPA, which seems logical (and valid), assuming they have less physical tasks in their job.

The LPTA seem all very high: 196-305 minutes are 3-5 hours a day! In the Netherlands a lot of people don't meet the recommended norm of 150 minutes moderate intensity per week. Insight into the categories would be helpful.

Mean JSS scores seem low (6.8 on a scale of 4-24) and not normally distributed, as indicated by the SD's. There are probably a small group of participants with high scores.

I would classify a correlation coefficient of 0.32 as low, not moderate.

As mentioned before, the analysis based on figure 2 is not described in Methods and the rationale is missing. Is this a good way to detect interaction? I have doubts about calculating correlations after first defining sub-groups. Please explain your rationale. What is the purpose of figure 2? Please interpret the correlation and its meaning. Fig.3 also is an uncommon way of presenting results, what is the purpose?

I notice gaps between the data analysis, the presentation of the results and the interpretation in the Discussion. Rationales should be better explained.

DISCUSSION

It is not clear what results the statements in the first paragraph are based on, please explain this specifically.

On page 13 the statement that 46% reported any insomnia symptoms 2-4 nights per week doesn't seem to match the data in table 1 (the average scores on the subscales around 2 indicate symptoms 1-3 nights/month).

	Also on the physical activity, new results are introduced in the Discussion, that are not easy to connect to the results in the tables. The other variables (Work ability, GHQ, etc.) are not discussed at all. Why is that, what was the purpose of measuring them? The reflections on page 14 should be more modest, regarding the low correlations found. The authors do come up with some possible reasons for results that do not support their hypothesis. Is a closer look at the characteristics of the group Low LPTA/high OPA perhaps necessary (lower education level, lower GHQ-12 and good health, job characteristics? Minor comment: could continuously wearing a waist-belt (also in the night) disturb sleep? CONCLUSION Of course sufficient sleep is important, but this study does not offer important, robust new insights. '... especially with ECEC professionals with a high level of PA during working days' : - but the subjects did not have vigorous PA at all (page 13, line27), so is this relevant? LITERATURE For several references (e.g. 6 and 36) authors are missing (only 1st author is mentioned).
--	---

VERSION 1 – AUTHOR RESPONSE

Reviewer: 1		
1	I suggest correcting the following words on line 18 of page 6: "understood" instead of "understanded" and "firstly" instead of "first".	Thank you for these notices. These words have been corrected. Please see p. 4, line 118.

Reviewer: 2		
1	This is very valuable work and the research questions are very important and there from the results very interesting. However, the hypothesis should be developed concerning high and low OPA.	Thank you for this comment. We understand the reviewer's point of view. In this study we are interested in both domains of physical activity, OPA and LTPA, and that is why our hypothesis is built on both PA domains. We feel that we cannot use terms high OPA and high LTPA in the hypothesis, since we have used these terms only to describe the amount of PA minutes per day and not included intensity. We have defined this in the text. Please see p. 6, lines 188-190. "For analysis, physical activity was defined as any level of physical activity (light, moderate, vigorous or very vigorous) and mean physical activity minutes/day at all PA intensity levels were added up. With the help of diaries, PA minutes were calculated separately for leisure time to represent LTPA and for occupational time to represent OPA. Later in this report high time (= high min/day) spent on LTPA or OPA will be referred as high LTPA or OPA."

2	Concerning the result section it needs to be less wordy, more structured and more informative e.g. picturing the results from Figure 1 and Table 1 respectively and then form a conclusion.	We have revised the Results section to make it less wordy. See for example p. 8, lines 249-251 and p. 9 line 261. Further, we restructured the section and moved the descriptive information of all participants to the beginning of the section. See p. 8, lines 242-245. Also, we added the following subheadings: “Association between recovery from work and insomnia in all participants”(p. 8, line 246) “Background characteristics” (p. 9, line 260) “Relationship between recovery from work and insomnia in four physical activity groups” (p. 11, line 284) “Relationship between recovery from work, physical activity and insomnia” (p. 12, line 298) Also, we have edited the text through the whole manuscript to refine the naming of the concept of recovery from work (earlier need for recovery after work) to better differentiate the phenomenon from its measure Need for Recovery (NFR). Hopefully these revisions make the Results section less wordy as well as more structured and informative.
3	The paragraph on Need for recovery should refer to table 1 etc.	Thank you for this observation. Please see above the answer above to the comment number 2 concerning the revisions in the result section. All references to figures and tables have been checked and corrected. We hope that these revisions made this part of results more informative.
4	The result section also does not deal with the hypothesis on insomnia, need for recovery, OPA and LTPA which is necessary.	Our hypothesis was that insomnia is associated with need for recovery after work, and that LTPA would enhance recovery and that OPA would deteriorate it. We have added subheadings to make it easier to follow the section and to show how it answers to our hypotheses. See also the response to the comment number 2 above explaining this. In the results section p. 9, lines 252-254 under new subheading (“Association between recovery from work and insomnia in all participants”) we present the results concerning the hypothesis of association between insomnia and recovery after work. Also, we think that information in Table 1. (= Table 2. in the original manuscript) answers to this question showing the correlations between the NFR and individual insomnia symptoms measured with the JSS. Results concerning the hypothesis of the role of OPA

		and LTPA within recovery after work and insomnia is presented in Results section under new subheading “Relationship between recovery from work and insomnia in four physical activity groups”. See p. 11, line 284. Figure 3 (= Figure 2. in the original manuscript) shows the result of age-adjusted correlations with 95% CIs between Need for Recovery after work and Jenkins Sleep Scale in four different physical activity groups based on low or high LTPA and OPA. See page 11, line 293. Finally, with Figure 4 (= Figure 3 in the original manuscript), we summarize the results of this study and present relationships between NFR and LTPA, OPA, age and JSS. See p. 12, line 303. Hopefully, the above-described revisions clarify the presentation of the results in line with our hypotheses.
5	The discussion also needs to be less wordy, more structured, selective, more distinct and more informative and include components regarding breaks and physical load during work.	We have revised the structure of the Discussion section to be more informative and to better follow the content of the Results section. We have revised the section to be less wordy, please see for example p. 12, lines 323-324; p. 13, lines 326-327 and p. 13, lines 345-352. Also, we have included components regarding work-related components of recovery to several parts of the section. Please see p. 13, lines 338-340; p. 15, lines 398-401.
6	The references should be presented so that they foremost could shed light on the current result.	We have revised the structure and order of the topics in the Discussion section to better present the references in the light of our results. Please see the whole Discussion section. We have also made a few revisions to the Discussion section by removing few references irrelevant to the results of this study. Please see p. 13, lines 338-340; p. 15, lines 398-401.
7	Concerning the conclusion there is room for some developments. Degrees of freedom concerning breaks at work or reasonably balanced demands at work are not mentioned but solely physical activity as the source of stress tolerance, fitness and health.	We have reformulated the conclusion paragraph and expanded the perspective as requested. Please see p. 15, lines 404-412.

	Reviewer: 3	
1	GENERAL: The physiological perspective (by measuring physical activity) on need for recovery and sleep in this study is interesting. However, I believe that the results should be interpreted with more constraints than the authors do, considering the low to moderate correlations	Thank you for the comment. We have reformulated the interpretation of the results, considering the the low to moderate correlations found. Please see: Abstract: p. 1, lines 29, 32 and 36-40 Results: p 9, lines 252-253 and p. 11, line 286

	found.	Conclusions: p. 15, lines 404-412.
2	ABSTRACT: In the conclusion the phrase 'insomnia and need for recovery are suggested to relate' is vague. The recommended 'attention' for phys.act. during free time is also vague and not supported by results.	We have reformulated the conclusion in the Abstract to better support the results of this study. Please see p. 1, lines 36-40
3	INTRODUCTION: Based on a good overview of relevant literature, the authors formulate a hypothesis for their study.	Thank you for the comment.
4	METHODS: The background of the DagsWork study is described; complete data of 224 participants were analysed. Please report brief information on psychometric aspects of the NFR.	Psychometric properties of the Need for Recovery after work scale has been presented in the following paper: de Croon et al 2006. Psychometric properties of the Need for Recovery after work scale: test-retest reliability and sensitivity to detect change. We added this reference to the text as well. Please see p. 5, lines 148-150.
5	Please report brief information on the thresholds for physical activity, what do the counts per minute express exactly?	When accelerometer measures physical activity, it counts the number of movements detected within a specific timeframe = counts per minute (cpm). Counts per minute refers to the number of times that an accelerometer detects movement (change in acceleration). The more counts, the higher is the intensity level of PA. Counts per minute is a well-established and commonly used way to report accelerometer-measured physical activity. In this study, based on earlier studies (for example Freedson et al 1998) all activity with 100 or more counts per minute was classified as PA ie. 100 cpm was used to separate sedentary time from PA time. Thresholds for sedentary time and physical activity and more specifically for each PA intensity levels used in this study are reported in Methods section in Physical activity paragraph, see page 6. lines 180-182. To be more precise, we revised the main text to explain that sedentary time was excluded from the accelerometer data. After this all PA at any intensity level was added up to represent PA. See p. 6, lines 183-184. Also, we added a short explanation of cpm in the main text, see p. 6, lines 182-183.
6	Please give brief explanation why these other variables (page 8, line 22) were selected and how they were included in analysis of data. They don't come back in the discussion.	Thank you for the comment. Other variables are included in the study because we wanted to describe the participants' backgrounds in different PA groups. These typical lifestyle variables and health variables are known to be related to our primary outcome variables, and thus describing the groups with these variables was felt to be important. The variables do not show any special

		differences in relation to OPA and LTPA, so they are not included in the discussion. However, we thank for the further comment concerning the Discussion section and have added consideration related to educational level to the Discussion section, see p. 14, lines 363-369.
7	Stat.analysis: what are FMI and LMI?	Thank you for this notice. There was a mistake in the manuscript. Abbreviations have been replaced with the correct terms. See p. 8, line 224.
8	In Methods, the calculation of correlations coefficients for the 4 combinations of OPTA and LPTA is not mentioned or explained; no rationale for this approach is given.	In Methods section the calculations of correlation coefficients is described as follows: Unadjusted and adjusted (partial) correlations were calculated by the Pearson method, using Sidak adjusted confidence interval. See p. 8, lines 232-233. The rationale is that we want to show whether the groups with different physical activity profiles differ in terms of how NFR and JSS are related. In other words, does physical activity profile have an association on the relationship between NFR and JSS. We revised the text to explain the rationale of this, please see p. 10, lines 271-272.
9	RESULTS Table 1: What calculation do the percentages in the row Highly educate indicate? 29 is 57% of what (57% of 53 is 30)?	It is because some participants had missing data in single variables.
10	The highly educated participants seem to have lower OPA, which seems logical (and valid), assuming they have less physical tasks in their job.	We agree with the reviewer's observation. In this study highly educated participants had less OPA. Reason for that is probably that even if all participants worked in ECEC centres, the highly educated participants usually have more administrative responsibilities compared to their less educated counterparts and that is reflected to the lower amount of physical activity at work.
11	The LPTA seem all very high: 196-305 minutes are 3-5 hours a day! In the Netherlands a lot of people don't meet the recommended norm of 150 minutes moderate intensity per week. Insight into the categories would be helpful.	Thank you for the comment. In the Methods section in Physical activity section, we reported that both OPA and LTPA measured in the study represent all physical activity intensity levels, including light PA as well: See page 6, lines 184-190: "For analysis, physical activity was defined as any level of physical activity (light, moderate, vigorous or very vigorous). Mean physical activity minutes/day at all PA intensity levels were added up. With the help of diaries, PA minutes were calculated separately for leisure time to represent LTPA and for occupational time to represent OPA. Later in this report high time (= high min/day) spent on LTPA or OPA will be referred as high LTPA or OPA." In this study participants' PA was mostly light in

		intensity. For these reasons LTPA is much higher (more minutes/day) than that of minutes of moderate intensity LTPA typically reported. The level of accelerometer measured PA in our study is very similar to for example Husu et al 2018: The objectively measured physical activity, sedentary behavior and physical fitness of Finns . They reported the amount of accelerometer measured PA per day as follows: 5.8 hours on light PA, 46 minutes on moderate PA, and 2 minutes on vigorous PA.
12	Mean JSS scores seem low (6.8 on a scale of 4-24) and not normally distributed, as indicated by the SD's. There are probably a small group of participants with high scores.	We agree with the reviewer's comment. The distribution is skewed to the right (mean 6,8; median 6,0), only 8 participants had score of 15 or more (max 18). JSS mean scores in our study are very similar to a large working population cohort in Finland (mean 6.4; median 4.8) (Juhola et al 2021: Internal consistency and factor structure of Jenkins Sleep Scale: cross-sectional cohort study among 80 000 adults). Their population was about the same age as our participants, and predominantly (82%) women.
13	I would classify a correlation coefficient of 0.32 as low, not moderate.	Thank you for the comment. We have reformulated the interpretation of the correlation coefficients according to the comment. Please see revisions: Abstract: p. 1, lines 29, 32 and 36-40 Results: p. 9, lines 252-253 and p. 11, line 286 Conclusions: p. 15, lines 404-412.
14	As mentioned before, the analysis based on figure 2 is not described in Methods and the rationale is missing. Is this a good way to detect interaction?	In figure 2 (= Figure 3. in the revised manuscript), we do not show interaction but correlation between NFR and JSS in four different PA groups. Further, please see answer to the question number 8.
15	I have doubts about calculating correlations after first defining sub-groups. Please explain your rationale.	According to the title, we are interested in the role of physical activity in connection with NFR and JSS. Therefore, we want to divide the participants into groups on the basis of physical activity and already in the initial phase show how physical activity is reflected in the characteristics of these different groups. We think this is important in interpreting the results in Figure 3. However, we have added subheadings and changed the order of the text in the Results section to make it easier to distinguish the results concerning all participants and results based on the four PA groups. Please see Reviewer 2, and the comment number 2.
16	What is the purpose of figure 2?	The purpose of the figure 2 (= Figure 3 in the revised manuscript) is explained above, please see the answer to the question number 8.

	Please interpret the correlation and its meaning.	Correlation between NFR and JSS is low to moderate in all PA groups. However, results indicate that the amount of OPA is more relevant than the amount of LTPA in relation between NFR and JSS.
17	Fig.3 also is an uncommon way of presenting results, what is the purpose?	Thank you for this question. With Figure 4 (= Figure 3. in the original manuscript), we summarize the results of this study. With the figure, we aim to describe the role of four different factors in explaining recovery and it enables us to present the strength of four different factors in predicting NFR. It allows us to compare the role of these four meters (operating on different scales) with each other in one figure. Previously, similar way of presenting results have been used for example in the following research articles:  • Karihtala et al. Relationship between occupational and leisure-time physical activity and the need for recovery after work Archives of Public Health (2023) 81:17, https://doi.org/10.1186/s13690-022-01017-8 • Munukka et al. Relationship between lower limb neuromuscular performance and bone strength in postmenopausal women with mild knee osteoarthritis. J Musculoskelet Neuronal Interact 2014; 14(4):418-424 • Päivärinne et al. Relations between subdomains of physical activity, sedentary lifestyle, and quality of life in young adult men. Scand J Med Sci Sports. 2018;28:1389–1396. https://doi.org/10.1111/sms.13003
18	I notice gaps between the data analysis, the presentation of the results and the interpretation in the Discussion. Rationales should be better explained.	Thank you for the comment. Results section is restructured (see for example see page X, lines xx-xx) and interpretation of the results have been revised in the Discussion section to better follow the results see page X, lines xx-xx. We hope these revisions fill in the gaps and rationales are now better explained.
19	DISCUSSION It is not clear what results the statements in the first paragraph are based on, please explain this specifically.	We have revised the first paragraph of the discussion to better explain what results the statements are based on. See p. 12, lines 309-316. Following sentence “Findings suggest that there is a relation between the need for recovery after work and both insomnia and physical activity during working hours.” (p. 12, lines 310-312)

		refers to the results based on results presented under the subheadings: Association between recovery from work and insomnia in all participants (p. 8, line 246), and Relationship between recovery from work, physical activity and insomnia (p. 12, line 298). The sentence “The results supported our hypothesis regarding the detrimental impact of occupational physical activity.. etc...” refers to the results under the subheading “Relationship between recovery from work and insomnia in four physical activity groups” Please see p. 11, line 284. We hope these revisions clarify our message.
20	On page 13 the statement that 46% reported any insomnia symptoms 2-4 nights per week doesn't seem to match the data in table 1 (the average scores on the subscales around 2 indicate symptoms 1–3 nights/month).	Thank you for bringing this to our attention. There was an error in the mentioned section in the manuscript. Our aim was to express that in this study, 46 percent of the participants experienced occasional symptoms of insomnia (any insomnia symptom occurring 1-4 times per month). We have made the necessary corrections to reflect this. See p. 12, line 322.
21	Also on the physical activity, new results are introduced in the Discussion, that are not easy to connect to the results in the tables.	Thank you, this is a useful comment. We have revised this part of the discussion section in order not to confuse with new results and to make it easier to connect with the results section of the manuscript. Please see p. 13, lines 345-352.
22	The other variables (Work ability, GHQ, etc.) are not discussed at all. Why is that, what was the purpose of measuring them?	The detailed background variables are included to describe the participants in detail in different PA groups. These typical lifestyle variables and health variables are known to be related to our primary outcome variables, and thus describing the groups with these variables was felt to be important. The variables did not show any special differences in relation to OPA and LTPA, except educational level and career history. However, we thank for the further comment concerning the Discussion section and have added there  - consideration related to career history and LTPA see p. 14, lines 364-369 - consideration related to educational level and OPA see p. 15, lines 393-394
23	The reflections on page 14 should be more modest, regarding the low correlations found.	Please see the answer to the comment number 1 and 13.
24	The authors do come up with some possible reasons for results that do not support their hypothesis. Is a closer look at the characteristics of the group Low LPTA/high OPA perhaps necessary (lower education	We have added discussion about the lower educational level to the manuscript. Please see p. 15, lines 393-394.

	level, lower GHQ-12 and good health, job characteristics?	
25	Minor comment: could continuously wearing a waist-belt (also in the night) disturb sleep?	Thank you for the comment. There was an error in the manuscript. During the nights the meter was moved from the waist to the wrist which is the golden standard for measuring sleep with accelerometer. We have added a mention of this into the manuscript, see p. 6, lines 173-174.
26	CONCLUSION Of course sufficient sleep is important, but this study does not offer important, robust new insights. '... especially with ECEC professionals with a high level of PA during working days' : - but the subjects did not have vigorous PA at all (page 13, line27), so is this relevant?	In this study high level of PA refers to high amount of minutes of PA in any intensity level and thus, it includes PA in light, moderate and vigorous intensities. We have revised this part of the Methods section to be more precise with this definition: Please see page 6, lines 184-190: "For analysis, physical activity was defined as any level of physical activity (light, moderate, vigorous or very vigorous) and mean physical activity minutes/day at all PA intensity levels were added up, With the help of diaries, PA minutes were calculated separately for leisure time to represent LTPA and for occupational time to represent OPA. Later in this report high time (= high min/day) spent on LTPA or OPA will be referred as high LTPA or OPA." Hence, our results suggest that high OPA (= high min/day), even in light intensity, ie. long-lasting, low-intensity OPA, may play a role in recovery.
27	LITERATURE For several references (e.g. 6 and 36) authors are missing (only 1st author is mentioned).	Thank you for this notice. References have been rechecked and missing authors have been added to the references.

VERSION 2 – REVIEW

REVIEWER	Bieleman, Andre Saxion University of Applied Sciences, Smart Health
REVIEW RETURNED	07-Feb-2024
GENERAL COMMENTS	I am satisfied with the authors thorough changes to the manuscript, addressing all reviewers' comments.